# Ultrasound Combination to Improve the Efficacy of Current Boron Neutron Capture Therapy for Head and Neck Cancer

**DOI:** 10.3390/cancers16152770

**Published:** 2024-08-05

**Authors:** Yoshiaki Yura, Yusei Fujita, Masakazu Hamada

**Affiliations:** 1Department of Oral & Maxillofacial Oncology and Surgery, Osaka University Graduate School of Dentistry, Suita, Osaka 565-0871, Japan; hamada.masakazu.dent@osaka-u.ac.jp; 2Department of Oral and Maxillofacial Surgery, Rinku General Medical Center, Izumisano, Osaka 598-8577, Japan; yusei.fujita@icloud.com

**Keywords:** boron neutron capture therapy, head and neck cancer, boronophenylalanine, ultrasound, microbubble, high-intensity focused ultrasound

## Abstract

**Simple Summary:**

Boron neutron capture therapy (BNCT) is radiotherapy that depends on a nuclear reaction between boron-10 (^10^B) and neutrons. The neutron source has been changing from nuclear reactors to accelerators. Current BNCT using boronophenylalanine (BPA) as the boron compound has been shown to markedly reduce tumor sizes when used to treat recurrent head and neck cancer (HNC); however, early recurrence has been reported. Ultrasound, a safe diagnostic method, in combination with microbubbles, is expected to promote the uptake of BPA into tumor cells. Ultrasound also has the ability to increase the sensitivity of cancer cells to radiotherapy. Therefore, the combination of ultrasound needs to be promoted in order to improve the antitumor efficacy of BNCT for HNC.

**Abstract:**

Boron neutron capture therapy (BNCT) is radiotherapy in which a nuclear reaction between boron-10 (^10^B) in tumor cells and neutrons produces alpha particles and recoiling ^7^Li nuclei with an extremely short range, leading to the destruction of the tumor cells. Although the neutron source has traditionally been a nuclear reactor, accelerators to generate neutron beams have been developed and commercialized. Therefore, this treatment will become more widespread. Recurrent head and neck cancer (HNC) close to the body surface is considered a candidate for BNCT using the boron compound boronophenylalanine (BPA) and has been found to be highly responsive to this treatment. However, some cases recur early after the completion of the treatment, which needs to be addressed. Ultrasound is a highly safe diagnostic method. Ultrasound with microbubbles is expected to promote the uptake of BPA into tumor cells. Ultrasound also has the ability to improve the sensitivity of tumor cells to radiotherapy. In addition, high-intensity focused ultrasound may improve the efficacy of BNCT via its thermal and mechanical effects. This review is not systematic but outlines the current status of BPA-based BNCT and proposes plans to reduce the recurrence rate of HNC after BNCT in combination with ultrasound.

## 1. Introduction

Boron neutron capture therapy (BNCT) is radiotherapy (RT) that destroys tumor cells by alpha particles and recoiling ^7^Li nuclei, which are produced by the ^10^B(n, α)^7^Li, boron neutron capture reaction, with high linear energy transfer values and very short flying distances of approximately 10 μM. Unlike conventional photon-based X-rays and gamma rays, BNCT selectively destroys tumor cells that accumulate boron compounds [1,2]. Clinical studies have examined the effects of BNCT on brain tumors and head and neck cancers (HNCs) that recurred after surgery and/or RT and are resistant to conventional chemoradiotherapy [3,4,5,6,7]. Since neutrons from nuclear reactors were the only source of neutrons, clinical studies on BNCT for HNC were limited to facilities in Finland, Japan, Taiwan, and other countries, and it did not become more widespread globally.

An important aspect of BNCT is the type of boron compound selectively transporting boron to tumor cells, which is central to this reaction. There have been extensive research efforts to develop boron compounds with tumor selectivity; however, boron compounds have proven to have low toxicity and use at the clinical level has been limited to boronophenylalanine (BPA) and sodium borocaptate (BSH) [8,9,10]. BSH is a compound containing 12 boron atoms. Although it has a high boron content, it diffuses into tumors in a concentration-dependent manner and is not tumor-selective. BPA was found to accumulate more in tumor cells that have an ability to synthesize protein actively using L-type amino acid transporter 1 (LAT1), a major transporter [11,12,13]. Since BPA is soluble in water when bound to fructose, it has to be prepared each time it is used. However, a new sorbitol formulation of BPA has been developed [14] and was approved for clinical use in Japan in 2022 as the sterile solution borofalan.

A recent major advance in BNCT is the development of an accelerator-based neutron generator that produces neutrons via the ^9^Be(p,n)^9^B or ^7^Li(p,n)^7^Be reaction [6,15,16,17]. Since accelerator-based BNCT systems can be installed in hospitals, BNCT is becoming more widespread as a global therapy. Many recurrent HNC have already been exposed to a significant dose of radiation in the initial treatment; therefore, re-irradiation is associated with a high risk of serious complications. HNCs are located relatively close to the body surface and remain deep enough to be reached by neutrons. Therefore, tumor-selective BNCT has been attracting attention as a treatment for recurrent HNC. BPA-based BNCT (BPA-BNCT) has been proven to be effective in the treatment of advanced or recurrent HNC using nuclear reactors. More cases of recurrent HNC will be treated with accelerator-based therapies in the future. In this review, we outline the status of BPA-BNCT for HNC using nuclear reactors and accelerators, the treatment of tumors with ultrasound, and treatment plans utilizing ultrasound in an attempt to improve the effectiveness of BPA-BNCT.

## 2. BNCT for the Treatment of HNC

The development of accelerators to generate neutrons increases the likelihood of clinical research with novel boron compounds for BNCT. However, the boron compound that has been used in current accelerator-based BNCT is limited to BPA. Therefore, data obtained by reactor-based BNCT for HNC may be used to evaluate clinical findings from accelerator-based BNCT.

Primary HNC is treated by surgery, radiation, and chemotherapy, depending on the site and degree of tumor progression [18,19,20]. In oral cancer, surgery is often performed with the aim of a radical cure, even in cases of T3 or T4, followed by reconstruction of the defect by flap transplantation. However, postoperative irradiation or radiochemotherapy is required if risk factors, such as the condition of resected margins or multiple cervical lymph node metastases, are confirmed in surgical specimens. In these cases, normal tissues surrounding the tumor are also exposed to a sufficient dose of radiation. Consequently, at the time of the recurrence of HNC, the lesion is often identified as a mass surrounded by scar tissue caused by surgery and/or photon-beam radiation therapy. Since BNCT is generally considered at the time of local recurrence, it is given a high hurdle to cure tumors from the outset of the treatment.

Many studies on BNCT for HNC have been conducted using nuclear reactors in Finland, Japan, and Taiwan (Table 1). The tumor-selective accumulation of BPA was demonstrated in HNC [21,22], and clinical findings on BPA-BNC for recurrent HNC [23,24,25] and accelerator-based BNCT for HNC have been reported [26]. Prior to BNCT, the accumulation of BPA in tumor and normal tissues was imaged and quantified as a tumor/blood ratio (T/B ratio) in a 4-borono-2-^18^F-fluoro-L-phenylalanine-fructose (^18^F-FBPA-Fr) positron emission tomography study. ^10^B concentrations in normal tissue were assumed to be equal to blood ^10^B concentrations during irradiation [27].

In Finland, BNCT was performed with neutrons from a nuclear reactor and BPA in two separate sessions; there was a 6-week interval between the first and second courses of irradiation [28]. This strategy is based on the assumption that second round of BNCT target cells are those that survived the first course of irradiation and acquired the proliferating capability to accumulate BPA. All 79 patients treated by Koivunoro et al. [29] had squamous cell carcinoma (SCC). The weighted total dose (Dw) was defined as the sum of the physical dose components (Di) multiplied by the weighting factors (wi) of each dose component in a tissue. The annotation “(W)” is added after the dose for clarity. The median minimum irradiation dose to the gross tumor volume (GTV) in the first round was 15 (12–18) Gy (W) and was 14 (10–16) Gy (W) in the second round. The rates of a complete response (CR), partial response (PR), stable disease, and progressive disease were 36, 32, 25, and 7%, respectively; the 2-year overall survival (OS) rate was 21%. It was not clear whether surviving tumor cells acquired the capability to take up BPA actively at the time of the second phase of BNCT.

BPA alone (72 patients) and a combination of BPA and BSH (13 patients) were used in a Japanese study on nuclear reactor-based BNCT for HNC [27]. BSH allows boron to reach tumor cells in a concentration gradient, and, thus, the addition of BSH is expected to increase the ^10^B concentration of cells in which the concentration of BPA does not reach a high level. Histologically, 33 (53%) SCC, 12 (19%) salivary gland carcinomas including mucoepidermoid carcinomas and adenoid cystic carcinomas, and 11 (18%) malignant melanomas were included. There were 13 (21%) first-onset cases and 49 (79%) cases of recurrence. The median minimum tumor dose was a 17.9 (4–44.5) photon-equivalent dose (Gy-Eq), and the median maximum doses to the skin and mucosa were 6.9 (2.7–30.3) Gy-Eq and 10.9 (4.4–17.2) Gy-Eq, respectively. The overall response rate for all patients was 58% within 6 months after BNCT and the median survival time was 10.1 months from the time of BNCT. One- and two-year OS rates were 43.1 and 24.2%, respectively. The difference between BPA alone and BPA + BSH was not clarified.

In Taiwan, BNCT was followed by X-ray therapy with image-guided intensity-modulated radiotherapy (IG-IMRT), based on the assumption that local control may be improved through the photon irradiation of a large field around recurrent GTV. [30]. Therefore, the BNCT dose was reduced, and the remaining dose was provided by IG-IMRT. Fourteen patients, including ten with SCC, three with other carcinoma, and one with sarcoma, had local recurrence. BPA was administered at 500 mg/kg. The median average BNCT dose for GTV was 21.6 (10.7–32.3) Gy-Eq and the median IG-IMRT dose for the planning target volume was 46.8 (41.4–53) Gy. The one-year OS rate was 56% and the progression-free survival rate was 21%. The response rate was high (64%); however, tumor regrowth was frequently detected. Recurrences were detected in or near (<1 cm) the radiation field in all cases; two patients had a recurrence on the contralateral side to the treatment field and one had both. There were no lymph node or distant metastases.

Findings on BNCT in Japan using an accelerator have been published (Table 1). The cyclotron-based neutron generator and borofalan were used to treat advanced and recurrent HNC patients (8 SCC and 13 non-SCC) [26]; 5 adenoid cystic carcinoma, 3 mucoepidermoid carcinoma, 2 acinic cell carcinoma, and 2 salivary gland conduit carcinoma, all of which were considered to be of a salivary gland origin, and 2 mucosal melanoma were included in non-SCC. BPA was administered at 400 mg/kg. The median tumor mean dose was 44.7 (42.9–50.6) Gy-Eq, and the median minimum dose was 31.1 (26.1–34.3) Gy-Eq. The overall response rate was 71%, with 24% CR and 48% PR at 90 days after BNCT. The two-year OS rates for SCC and non-SCC were 58 and 100%, respectively. This study also described the treatment regimen for recurrence after BNCT. During the follow-up for SCC, one patient was treated with RT, one with chemotherapy, and three with immune checkpoint inhibitors (ICIs), indicating that 50% received additional treatment after BNCT. Regarding non-SCC, 11/13 patients (85%) received treatment after BNCT; eight were treated with surgery and postoperative radiation, one with surgery, one with radiochemistry, and one with RT. Many non-SCC cases were withdrawn from the study because patients who refused surgery were added to the study and no restrictions were placed on treatment beyond 3 months of BNCT.

## 3. Current BPA-Based BNCT Issues

Previous studies on BPA-BNCT for HNC reported a high response rate and rapid reductions in tumor volumes without severe damage to surrounding normal tissue; however, local control was not necessarily sustained in the long-term and 1- and 2-year survival rates decreased [27,28,30]. This may be due to the diversity of tumor cells. For example, tumor cells that are in the quiescence phase, express low levels of LAT1, are located far from capillaries, may not actively take up BPA, and the radiation dose by BNCT may not reach lethal levels [31]. In addition, a study with epithermal neutrons from nuclear reactors showed that the maximum tumor depth ranged from 0 to 100 mm, and the minimum tumor dose from 4.0 to 44.5 Gy-Eq [29], suggesting that the dose delivered to deep-seated tumors was insufficient to control tumors. This limitation needs to be considered in the planning of BNCT for HNC.

The T/N ratio is considered to be >3/1, which means that normal tissue in the neutron irradiated field may also be exposed to one-third of the radiation dose delivered to the tumor [32]. Furthermore, the ^10^B(n, α)Li^7^ nuclear reaction, in which ^10^B captures low-energy (<0.5 eV) thermal neutrons, produces not only alpha particles (1.47 MeV) and recoiling ^7^Li nuclei (0.84 MeV), but also contaminating long-flying gamma rays (0.48 MeV) [33]. The other main causes of doses to healthy tissues are (at depth) neutron capture in nitrogen and gamma rays from neutron capture in hydrogen. Therefore, it is inevitable that BNCT for HNC induces some damage to the surrounding normal tissues, followed by complications, such as mucositis, dermatitis, osteonecrosis of the jaw, meningitis, carotid artery injury, hyperamylasemia, and alopecia [27,34,35,36]. Therefore, the distance from tumors to critical tissues needs to be rigorously evaluated.

## 4. Ultrasound as a Strategy to Enhance the Effectiveness of BPA-Based BNCT

To prevent the recurrence of HNC after BPA-BNCT, non-radioactive treatment modalities need to be applied in combination with BNCT. One approach to overcome the heterogeneity of boron concentrations in tumor cells is to forcibly transport boron compounds into tumor cells by electroporation [37,38]. However, it has not yet reached the clinical application stage due to its invasive nature. In contrast, ultrasound is less invasive and has already been used for the diagnostic imaging and treatment of diseases such as bone fracture and neoplastic diseases. More importantly, ultrasound may be used to enhance drug delivery and decrease cell viability through its thermal and mechanical effects. Therefore, ultrasound has the potential to improve the therapeutic effects of BNCT for HNC.

### 4.1. How Ultrasound Works and Its Application to Cancer Treatment

Important parameters in ultrasound irradiation are its frequency, special peak temporal average intensity (ISPTA), sound pressure, pulse repetition rate, duty cycle (DC), and irradiation time [39]. In vivo, the absorption coefficient increases in proportion to the ultrasound frequency; therefore, the transmission rate decreases with increases in frequency. Cavitation is a phenomenon in which ultrasound causes bubble formation and vibrations in a liquid and is a fundamental mechanism for therapeutic ultrasound [40,41]. Ultrasound intensity is the ultrasonic energy flow per unit area, which is generally expressed in W/cm^2^. Sound pressure (Pa) is the pressure change that occurs as ultrasound passes through a medium [39]. In this connection, Xin et al. [42] divided therapeutic ultrasound into two groups; low-intensity ultrasound (<3 W/cm^2^) and high-intensity ultrasound (≧3 W/cm^2^) (Figure 1A). When ultrasound exerts a thermal effect, the increase in the temperature of biological tissue is defined by factors such as intensity, frequency, the ultrasound absorption coefficient of the tissue, the heat capacity of the tissue, and the ultrasound exposure time. Ultrasound also exerts a mechanical effect when its sound pressure increases to induce cavitation in the tissue. The mechanical index (MI) is defined by the rarefractional pressure inside the tissue and frequency. If MI is less than 1, the likelihood of cavitation is considered to be low. At low frequencies and high-pressure amplitudes, mechanical effects may be readily induced, while thermal effects predominate at higher frequencies and longer pulses. Low-intensity ultrasound is used in imaging medicine (imaging, diagnostics, and drug delivery), while high intensity is employed in treatments such as ablation and surgery. Furthermore, in terms of frequency, a low frequency (20–200 kHz) is used for drug delivery, surgery, and ablation, a moderate frequency (0.7–3.0 MHz) for accelerated fracture healing, wound healing, and inflammation relief, and a high frequency (1–20 MHz) for imaging medicine and diagnostics (Figure 1B) [42].

High-intensity focused ultrasound (HIFU) was developed to limit the ultrasound exposure field. It induces localized thermal and mechanical effects due to compressions and decompressions induced by acoustic pressure waves applied to tissues [39]. Since its antitumor effect is limited to the area of tumors, ultrasound exposure is repeated to cover the total tumor volume. When focused ultrasound (FUS) is applied in its thermal mode, the tissue is subjected to frictional forces that are converted to heat; heating to >56 °C induces thermal necrosis and cell death. Histotripsy is the term used to describe FUS exposure consisting of a high-amplitude peak negative pressure (>15 MPa) for milliseconds or less to induce mechanical damage without temperature elevations. Pulsed HIFU exposure may be used to permeabilize and/or destroy tissue using a lower acoustic pressure (<10 MPa) and longer exposure time (> milliseconds) than histotripsy.

### 4.2. Research on Ultrasound Combined with Microbubbles (MBs) to Improve the Delivery of Drugs to Tumors

Since MBs are already used as contrast agents in ultrasound imaging, commercially available MBs may be used in cancer therapy. The shells that coat MBs are lipids, proteins, and polymers, and typical nuclear gasses are air, nitrogen, and perfluorocarbon [43,44,45]. When administered MBs are exposed to ultrasound, they begin to oscillate and undergo expansion and contraction. Stable cavitation (non-inertial inertia) occurs when a low acoustic pressure is applied, producing liquid flow in the vicinity of the cell membrane. Inertial cavitation is characterized by oscillations at a higher acoustic pressure, causing the contraction and expansion of bubbles and consequently their rupture [46,47,48]. The exposure of MBs to ultrasound, even if its intensity or acoustic pressure is low, induces cavitation and increases the permeability of cell membranes through the formation of small pores, called sonoporation. In actual cancer treatment, ultrasound with MBs (USMBs) has been shown to increase the concentration of chemotherapeutic drugs in tumor cells and improve their anti-tumor capability [44,49,50] (Table 2) (Figure 2A).

In human pancreatic cancers in nude mice, nanodroplets containing paclitaxel were administered intravenously under MRI, and within 6 h FUS with an ultrasound pressure of 3.1 MPa was performed. This treatment decreased tumor volumes or resulted in the complete loss of tumors. Anti-tumor effects were not observed when FUS was performed before or immediately after drug administration [51].The combination of gemcitabine with USMB was evaluated in inoperable pancreatic cancer patients and the findings obtained showed that it did not result in additional toxicity over that of gemcitabine chemotherapy alone. Median survival increased from 8.9 months to 17.6 months [52]. Interstitial tissue fluid pressure (IFP) and collagen fibers as an extracellular matrix inhibit the intracellular spread of nanoparticles in the tumor microenvironment (TME). To overcome this inhibition, nanoparticles were administered systemically, and tumor tissues were treated with non-destructive pulsed FUS. Consequently, more nanoparticles were delivered into tumors, suggesting that pulsed FUS changed the cytoarchitecture of the TME [53]. In human pancreatic cancer in nude mice, gemcitabine and paclitaxel were administered intravenously and tumors were then treated with USMB. Greater reductions in tumor volumes were observed when ultrasound was performed at a higher ultrasound power [54]. Ten patients with HER2-negative breast cancer were treated with USMB to enhance the effects of neoadjuvant chemotherapy (NAC) consisting of taxane, anthracycline, and cyclophosphamide. Within 24 h after the infusion of drugs, patients received USMB in each cycle of chemotherapy. Although the combined treatment group showed a higher response and clinical benefit rates than those in the NAC alone group, no significant difference was noted in the Response Evaluation Criteria in Solid Tumors (RECIST1.1) [55]. The effects of FUS and MBs were also investigated in 17 patients with liver metastasis of colorectal cancer. Two lesions were selected in each patient’s liver and treated with USMB after chemotherapy; the regimen was FOLFIRI (calcium folinate, fluorouracil, irinotecan) or FOLFOXIRI (calcium folinate, fluorouracil, oxaliplatin, and irinotecan). Although lager metastatic lesions were more likely to be reduced by USMB than control lesions, mixed responses to chemotherapy and the heterogeneity of lesions made it difficult to interpret the findings obtained [56].

### 4.3. Research on Improving the Effectiveness of RT with Ultrasound 

Cavitation by ultrasound may affect the integrity of endothelial cells as well as tumor cells by opening cell–cell junctions and leaking transport molecules from blood vessels into the TEM [47]. This may potentiate the anti-tumor effects of RT (Table 2). Human breast cancers on nude mice were treated with USMB. Thereafter, tumors received 0.2 or 0.8 Gy of X-ray irradiation. Mice were sacrificed 12 and 24 h after treatment for a histopathological examination and tumor growth delay was assessed 28 days after treatment. When combined with radiation, USMB acted as an enhancing agent and elicited synergistic anti-tumor and anti-vascular effects [57]. In a nude rat model of hepatocellular carcinoma (HCC), liver tumors received 5 Gy of X-ray irradiation alone or in combination with USMB. By day 7, the normalized tumor volume in the USMB + radiation group was 31% that of the of USMB group and 59% that of the radiation group. Survival was significantly prolonged when RT was combined with USMB. Therefore, USMB may be an effective adjuvant in the treatment of HCC by RT [58]. In nude mouse tumors of human nasopharyngeal carcinoma, USMB mildly decreased blood flow and CD34 expression, but enhanced tumor cell death by radiation [59]. A previous study investigated whether short FUS-Cav, which induces cavitation, affected the anti-tumor effects of X-ray irradiation or hyperthermia (45 °C, 30 min) on HNC, glioblastomas, and prostate cancer. Short FUS-Cav itself did not injure cancer cells, but increased the sensitivity of tumor cells to X-ray irradiation or hyperthermia [60]. The effects of pulsed HIFU in combination with RT were also examined in nude mouse tumors of prostate cancer. Ultrasound was performed at 1 MHz, and its intensity was 25 W, covering a significant margin of the tumor. Pulsed HIFU was found to enhance the effects of radiation on prostate cancer [61].

Sonoporation has been shown to induce apoptosis, arrests the cell cycle, and inhibits the repair of DNA damage [67]. Nofiele et al. [68] examined USMB-induced changes in ceramide levels using human cells including endothelial cells, acute myeloid leukemia cells, mouse fibrosarcoma cells, prostate cancer cells, breast cancer cells, and astrocytoma cells and found that ceramide production increased and cell viability decreased in all cell lines tested; the decrease in cell viability was more pronounced in vascular endothelial cells. The combination of USMB and radiation further increased ceramide production in endothelial cells and decreased cellular viability, while a decrease in cell viability was prevented by the addition of sphingosine-1-phosphate (S1P) to the culture or in acid sphingomyelinase (ASMase)-deficient mice. These findings suggest that USMB stimulates ASMase activity and ceramide production, leading to the radiosensitization of cells. Hyperthermia inhibits the repair of RT-induced DNA damage. It suppresses homologous recombination and leads to the shunting of early DNA double-strand break repair to non-homologous end rejoining, which is more error-prone [69,70]. The hyperthermic effects of ultrasound may increase the sensitivity of tumor cells to RT.

### 4.4. Research on the Use of FUS to Shrink Tumors 

In thermal ablation with HIFU, temperatures need to be increased to >56 °C; thermal necrosis and instantaneous cell death occur under this condition [71,72,73]. On the other hand, when the high negative pressure of HIFU is applied to tissue, tissue damage occurs due to the mechanical effects of ultrasound without temperature elevations [74,75] (Table 2) (Figure 2B).

The thermal effects of ultrasound on HNC have been examined in mouse cancer tissue and muscle. When exposed to HIFU in the continuous mode, necrosis by ablation was confirmed in tumor and muscle tissues [62]. Clinically, benign thyroid nodules were treated with HIFU to decrease tumor volumes by thermal ablation. The uptake of ^99m^Tc-MIBI or ^99m^Tc-pertechnetate was recorded pre- and post-treatment. After a heat treatment at approximately 85 °C, the median reduction in ^99m^Tc-MIBI uptake was 35.5%, while ^99m^Tc-pertechnetate scintigraphy revealed that the median reduction in uptake was 27% [63]. The application of FUS to treat a large oral tumor, a neurilemmoma measuring 4.3 × 3.8 × 3.8 cm in the maxilla of a dog, was reported. Three FUS treatments, each covering 50% of the tumor’s nuclear area, were performed over 1 month. Complete tumor regression was observed. Mucosal burns were noted as a treatment-related adverse event; however, the exposed bones were covered with gingival flaps [66].

The effects of non-ablative pulsed HIFU were also studied in melanoma and breast cancers in syngeneic mice. The parameter of pulsed HIFU used was a peak negative pressure of 6 MPa. The findings obtained showed a significant decrease in tumor diameters after the ultrasound treatment, but not complete disappearance [64]. In a study on pancreatic cancer in syngeneic mice, tumors were exposed to pulsed HIFU. ICIs, anti-CTLA-4/PD-1 antibodies, were also used in combination with pulsed HIFU. The treatment with pulsed HIFU resulted in detectable acoustic cavitation. CD8^+^ T-cell infiltration into tumors was more pronounced in the pulsed HIFU and pulsed HIFU + ICIs groups than in the sham-exposed control group. The survival of animals treated with pulsed HIFU + ICIs was also longer than that in untreated controls and those treated with either pulsed HIFU or ICIs alone. The mechanical action of pulsed HIFU was suggested to increase the infiltration of immune cells into the TME and create a proinflammatory environment to support the effects of ICIs [65].

## 5. Previous BNCT Studies Using Ultrasound

In BNCT for brain tumors, USMB was examined to establish whether it transiently disrupted the blood–brain barrier (BBB) and enhanced the delivery of BPA into the brain (Table 3). 

Rat brain 9L gliosarcomas were exposed to FUS with MBs. The peak negative sound pressure was 0.6 MPa. BPA–fructose (BPA-Fr) was administered intravenously at a dose of 250 mg/kg; 25% of the total dose was delivered as a bolus, followed by a single sonication to disrupt the BBB and delivery of the remaining BPA-Fr over a 2 h infusion. This treatment increased the uptake and accumulation of ^10^B in the main tumor, with a tumor-to-brain tissue ratio of 6.7 ± 0.5 with FUS with MBs and 4.1 ± 0.4 in the control group [76]. To estimate the uptake of BPA into F98 gliomas in the brain, rats received an intravenous injection of ^18^F-FBPA-Fr with or without USMB and its pharmacokinetics were analyzed by scanning for approximately 3 h. The tumor-to-contralateral brain ratio was approximately 1.75-fold higher for USMB-treated tumors than for control tumors. [78]. Polyethylene glycol-b-poly ((closo-dodecaboranyl) thiomethylstyrene) (PEG-b-PMBSH) nanoparticles coupled with cationic MBs (B-MBs) were used in combination with FUS to disrupt the BBB. T/N and T/B ratios were 3- and 2.3-fold higher, respectively, in the B-MBs group than in the PEG-b-PMBSH and MBs mixture group, indicating that B-MBs safely disrupted the BBB and enhanced boron delivery into tumor tissue [80].

In the head and neck region, the effects of USMB on the intracellular uptake of BPA were examined using human oral cancer cells. Cellular ^10^B concentrations increased after the incubation of cells with BPA, but not with BSH. When tumor cells were cultured with BPA or BSH and then subjected to USMB, intracellular ^10^B concentrations were significantly higher than those in untreated controls. The suppressive effects of BPA-mediated BNCT on cell viability were more prominent with those than without ultrasound [79]. In another study on nude mouse tumors of human tongue cancer, ^18^F-FBPA-Fr was intravenously injected immediately after pulsed HIFU, and microPET imaging and a biodistribution analysis were performed, which revealed the high tumor uptake of ^18^F-FBPA-Fr in mice treated with pulsed HIFU [77].

## 6. Proposed Plans to Reduce the Recurrence of BPA-BNCT-Treated HNC by Ultrasound

Based on the effects of ultrasound described in previous sections, several strategies to enhance the efficacy of BPA-BNCT for HNC by ultrasound need to be considered.

### 6.1. Increases in ^10^B Concentrations in Tumors and Endothelial Cells by USMB

USMB may be used to enhance the uptake of BPA into tumor cells. In the treatment of recurrent HNC, BPA is continuously injected intravenously to maintain the blood boron concentration and USMB is then performed in an area covering the GTV. It is critical to understand whether the application of ultrasound causes additional boron accumulation in cells that previously accumulated boron or in cells that did not actively accumulate boron previously. Further in vitro and in vivo studies are needed to clarify this important point, but a study using cultured oral cancer cells and MBs showed that ultrasound further increased the level of boron taken up by the cells [79]. USMB is expected to elevate the intracellular concentration of ^10^B, even in tumor cells in which the uptake of ^10^B does not spontaneously reach a high level (>20 μg ^10^B) [32,81]. In addition, BPA uptake by endothelial cells in tumor vessels as well as tumor cells may be enhanced by sonoporation. When ^10^B-containing endothelial cells are exposed to neutrons, endothelial cells are damaged by alpha particles and recoiling 7Li nuclei, leading to their destruction. In addition, ceramide, produced by ultrasound and RT, will contribute to the damage of endothelial cells. Incidentally, when nude mouse HNC tumors were histologically examined, damage to endothelial cells was observed one or two days after BPA-BNCT [22]. USMB is also expected to inhibit the effects of IFP and collagen fibers on the delivery of drugs and nanoparticles in the TME (Figure 3A).

The tumor vascular network allows for the transport of nutrients, oxygen, and immune cells and is regulated by pro- and anti-angiogenesis factors [82]. Vascular endothelial growth factor (VEGF) is an essential factor that promotes angiogenesis; related family factors and their receptor, VEGF receptors, have also been identified. Although the combination of USMB with BPA-BNCT may destroy the endothelial cells of tumor blood vessels and block the supply of nutrients and oxygen to surviving tumor cells, angiogenesis will promptly occur after BNCT though the effects of many angiogenic factors provided by neighboring normal tissues. Under these conditions, post-BNCT therapies targeting angiogenesis are useful for preventing the repopulation of surviving tumor cells and recurrence.

### 6.2. Prevention of the Repopulation of Surviving Cells after BPA-BNCT by HIFU

Recurrent lesions of HNC generally form three-dimensional tumor masses encapsulated within fibrous scar tissue or graft tissue. These are infiltrated by a meshwork of small blood vessels from the surrounding normal tissue. Burgos-Panadero et al. [83] divided tumor masses into three regions in the following order from the outside: proliferative, quiescent, and necrotic regions. When recurrent HNC lesions become larger, these regions appear and cells in quiescent regions adjacent to necrosis are suspected to have low levels of BPA and be resistant to BPA-BNCT.

In the head and neck region, tumors including benign thyroid tumors in patients and oral neurilemmoma in animals have been treated with FUS via its thermal effects. Therefore, if BPA-BNCT-resistant tumor cells are suspected to be present in the center of the tumor mass, HIFU represents an option for reducing cell viability through its thermal effects. HIFU in the continuous ablation mode for several seconds increases the local temperature of the targeted tissue to >56 °C, activating the pathways of programmed cell death and necrosis. However, when the range of heating is extended over the border of the tumor, damage to normal tissues occurs. Therefore, the control of tumor cells at the tumor periphery is a challenge for thermal ablation by HIFU (Figure 3B).

Pulsed HIFU has a high acoustic pressure to produce cavitation with exposure times lasting milliseconds or less and it damages tissues though mechanical action without destroying cells or temperature increases. The peak negative sound pressure of pulsed HIFU needs to be lower than histotripsy parameters (P = 17 MPa, DC = 1%, and 10 msec) in order to avoid non-specific inflammation in surrounding normal tissues. Even if tumor cells with insufficient amounts of ^10^B survive DNA damage by BPA-BNCT, the addition of pulsed HIFU may lead to cell death and tumor regression through its mechanical action, the inhibition of DNA repair, and increases in ceramide production (Figure 3B). In this case, pulsed HIFU may cover the extent of GTV, including the margin of tumors. Another option is to combine the treatment of plans A and B, performing A and then pulsed HIFU in B.

## 7. Future Prospectives

A number of new boron compounds have been developed. The properties required of these compounds are low toxicity, high uptake in tumors but low concentrations in normal tissues, and fast decay from blood and normal tissues but persistence in tumors. They range from small-molecular-weight compounds to boron-containing complexes modified with glucose, folic acid, amino acids, cell-permeable peptides, nucleoside and carbohydrate analogs, nucleotide borate esters, DNA intercalators, metallocaboranes, porphyrins, boron-charged antibodies, liposomes, and nanoparticles [32,33,81,84,85]. Some boron compounds are based on BPA and PSH; they include BPA-amide alkyl dodecaborate [86], poly(vinyl alcohol)-BPA [85], BSH-Z33 peptide-cetuximab targeting epidermal growth factor receptor (EGFR) [87], and poly-arginine peptide conjugated with BSH [88]. Other new boron compounds include, for example, cyclic RGD functionalized closo-dodecaborate albumin conjugates [89], ^10^B-enriched ^10^BPO_4_ nanoparticles surface-modified with an anti-EGFR antibody [90], boron carbide nanoparticles [91,92], pH-sensitive poly lactic-co-glycolic nanoparticles [93], and high boron-loaded nucleic acids [94].

Due to their size, MBs remain within blood vessels, making it difficult to exert therapeutic effects in area distant from blood vessels. In contrast, nanobubbles take advantage of the enhanced permeability and retention effects of tumor blood vessels to migrate outside the vessels and reside between tumor cells. They may serve as the nucleus for cavitation by ultrasound and may increase the cell permeability of blood vessels and tumor cells, contributing to an overall increase in the intracellular concentration of ^10^B [95,96]. Neutron irradiation under these conditions may result in more intense irradiation and cellular injury. Since nanobubbles have already been used at the research level, their clinical application is expected.

In hypoxic cells, the uptake of BPA is inhibited because hypoxia-inducible factor-1 alpha (HIF-1α) decreases the expression of LAT1. Therefore, YC-1, a HIF-1α inhibitor, may enhance the antitumor effects of BNCT by increasing boron concentrations, even in hypoxic tumor cells [97]. In addition, the expression of EGFR and *TP53* mutations have been identified as the most frequent genetic alterations in HNC, occurring at frequencies of 95 and 75–85%, respectively, in non-HPV-related tumors [98]. Furthermore, *TP53* mutations have been associated with decreased patient survival. In an animal study on BPA-BNCT, tumors of oral SCC cells with *TP53* mutations showed shrinkage upon BNCT, similar to cells with wild-type *TP53*; however, after a period of time, repopulation occurred in *TP53-*mutated cells only, suggesting the involvement of *TP53* mutations in the recurrence of HNC after BPA-BNCT [99]. Therefore, a treatment that restores p53 function may be effective. A number of compounds have been shown to directly affect mutated p53 by binding to it and restoring the wild-type conformation and transcriptional activity, including COTI-2, CP-31398, PRIMA-1, and APR-246 [100]. COTI-2 was found to synergistically increases the efficacy of cisplatin and RT in the treatment of HNC, both in vitro and in vivo [101]. Therefore, the restoration of p53 function with these drugs is expected to increase the efficacy of BNCT. The combined effects of ICIs and radiation have recently been reported [65]. Therefore, further research needs to be conducted on the combination of ICIs and BNCT.

## 8. Conclusions

In recurrent HNC, the surrounding tissues have already been exposed to radiation, and re-irradiation increases the risk of overexposure and severe complications. Since tumor cells with growth potential and expressing LAT1 actively take up BPA, BPA-based BNCT has potential as a selective radiation therapy, sparing the surrounding normal cells. A significant reduction in tumor volumes is generally observed following BPA-BNCT with less adverse events. Accelerator-based BNCT has been attracting increasing attention as a treatment modality for recurrent HNC. However, in some cases, tumor cells survive BNCT, proliferate and form recurrent lesions. When the risk of recurrence is high, USMB may be combined with BPA-BNCT to increase the concentration of boron in the tumor and vascular endothelial cells, thereby enhancing the death of these cells. Alternatively, after BPA-BNCT, the tumor may be cauterized with HIFU or damaged by the mechanical action of pulsed HIFU. Since the ultrasound contrast agents and equipment needed for HIFU are commercially available, ultrasound may be immediately applied to BNCT. It is also desirable to develop BPA-BNCT in combination with other systemic therapies such as inhibitors of angiogenesis, inhibitors of ceramide-metabolizing enzymes, and ICIs, and to apply new boron compounds and nanobubbles to clinical settings.

## Figures and Tables

**Figure 1 cancers-16-02770-f001:**
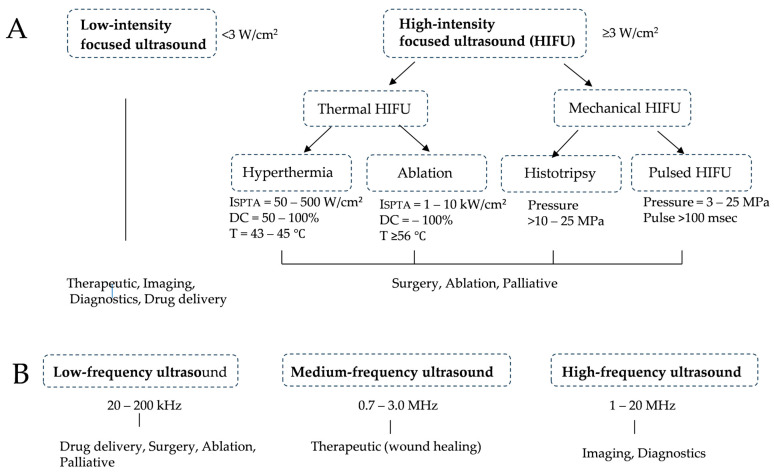
Outlines of therapeutic ultrasound. (**A**) Based on its intensity, focused ultrasound is divided into low- and high-intensity ultrasound. (**B**) Based on its frequency, ultrasound is classified into low-, medium-, and high-frequency ultrasound. DC, duty cycle. ISPTA, special peak temporal average intensity.

**Figure 2 cancers-16-02770-f002:**
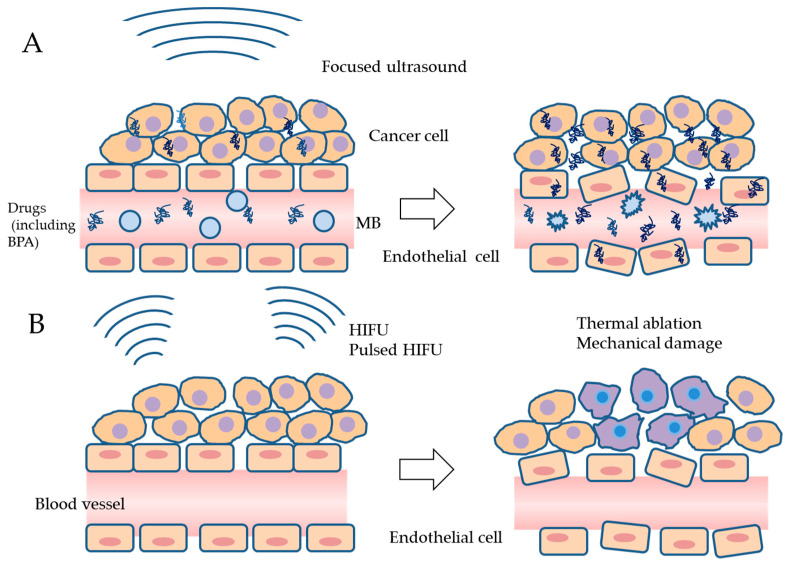
Effects of USMB and HIFU on drug delivery and cell integrity. (**A**) After the intravenous administration of drugs (include BPA) and microbubbles, ultrasound induces microbubble cavitation and enhances drug delivery into tumor cells and endothelial cells. (**B**) HIFU may induce the thermal ablation of tumors at temperatures >56 °C. Alternatively, the mechanical action of pulsed HIFU with a high negative pressure may damage tumor tissues without temperature elevations. MB, microbubble; USMB, ultrasound with MB; BPA, boronophenylalanine; HIFU, high-intensity focused ultrasound.

**Figure 3 cancers-16-02770-f003:**
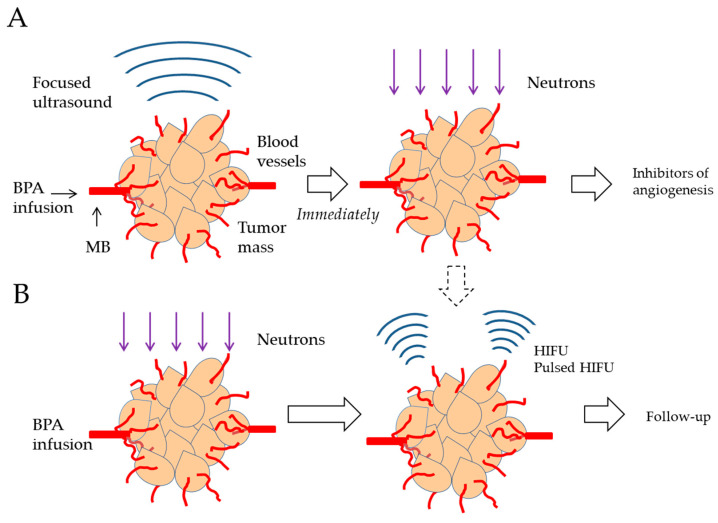
Focused ultrasound to prevent the recurrence of BPA-BNCT-treated HNC. (**A**) After the intravenous administration of BPA, USMB is performed in an area covering the GTV and to increase the concentration of ^10^B in tumor cells, followed by immediate neutron irradiation. USMB promotes the effects of BPA-BNCT on tumor cells and vascular endothelial cells. A molecular therapy that targets angiogenesis may prevent the repopulation of endothelial cells and tumor regrowth. (**B**) Recurrent tumors are treated with BPA-BNCT and then exposed to thermal ablation with HIFU or pulsed HIFU to injure surviving tumor cells after BNCT. (**A**) and (**B**) may be combined. BPA, boronophenylalanine; MB, microbubble; HIFU, high-intensity focused ultrasound.

**Table 1 cancers-16-02770-t001:** BNCT for the treatment of head and neck cancer.

References Year	Number of Patients, Histology	Boron Delivery Agents and Administration	Neutron Source and Dose	Clinical Outcome
[28] 2012	30 patients with recurrent malignancies, 29 evaluable SCC = 24 Sarcoma = 1	BPA Once = 4 Twice = 26	Reactor First average weight GTV dose = 23 (14–37) Gy (W) Second average weight GTV dose = 22 (15–30) Gy (W)	ORR = 76% MST = 13.0 months 2-year PFS = 20%, OS = 30%
[27] 2014	62 patients with advanced or recurrent cancers SCC = 33 Mucoepidermoid Ca = 5 Adenoid cystic Ca = 4	BSH + BPA or BPA (BPA = 72, BSH + BPA = 13) Once = 42 Twice = 17 Three times = 2 Five times = 1	Reactor Minimum tumor dose = 17.9 (4.0–44.5) Gy-Eq	ORR = 58% MST = 10.1 months 1-year OS = 43.1% 2-year OS = 24.2%
[29] 2019	79 patients with recurrent SCC, 69 evaluable	BPA Once = 40 Twice = 39	Reactor First median minimum GTV dose = 15 (12–18) Gy (W) Second median minimum GTV dose = 14 (10–16) Gy (W)	ORR = 68% OS = 21%
[26] 2021	21 patients with recurrent or locally advanced malignancies SCC = 8, Non-SCC = 13 (5 adenoid cystic Ca, 3 mucoepidermoid Ca, 2 acinic cell Ca, 2 salivary ductal Ca, 1 melanoma)	BPA	Cyclotron-based epithermal neutrons Median tumor mean dose =44.7 (42.9–50.6) Gy-Eq Median tumor minimum dose = 31.1 (26.1–34.3) Gy-Eq	ORR = 71% SCC: CR = 50%, PR = 25% 2-year OS = 58%, Re-reated with RT, Chem, ICIs Non-SCC: CR = 8%, PR = 62%, 2-year OS =100% Re-treated with surgery, RT, CRT
[30] 2023	14 patients with locally recurrent malignancies SCC=10 Mucoepidermoid Ca = 1 Non-keratinizing Ca = 1 Sarcoma = 1	BPA	Reactor Average GTV dose = 21.6 (10.7–32.3) Gy-Eq Fractionated IG-IMRT Total prescription dose = 46.8 (41.4–53) Gy	ORR = 64% 1-year PFS = 25%, OS = 56%

BPA, boronophenylalanine; BSH, sodium borocaptate; Gy-Eq, Gy equivalent; ORR, objective response rate; MST, median survival time; GTV, gross tumor volume; PFS, progression-free survival; OS, overall survival; Ca, carcinoma; CR, complete response; PR, partial response; RT, radiotherapy; CRT, chemoradiotherapy.

**Table 2 cancers-16-02770-t002:** Ultrasound in cancer research.

**Ultrasound with MB to Improve Drug Delivery**					
**References Year**	**Cancer**	**Purpose**	**Type of** **Ultrasound**	**Parameters**	**Effects**
[51] 2013	Human pancreatic cancer, nude mouse tumor	Delivery of paclitaxel-nanoparticles with MB	FUS	Frequency = 3 MHz, Acoustic pressure = 3.1 MPa 16-element annular transducer	Decreased tumor growth
[52] 2016	10 patients with inoperable pancreatic cancer	Delivery of gemcitabine with MB	FUS	Frequency = 1.9 MHz Intensity = 0.25 mW/cm^2^, MI = 0.2 (peak negative pressure = 0.27 MPa), DC = 0.3%	May improve the clinical effects of gemcitabine
[53] 2021	Human hypopharyngeal cancer, nude mouse tumor	Delivery of nanoparticles with MB	Pulsed FUS	Frequency = 500 kHz ISPTA = 1.55 kW/cm^2^ Acoustic pressure = 5 MPa	Improved nanoparticle delivery
[54] 2021	Human pancreatic cancer, nude mouse tumor	Delivery of abraxane IV and gemcitabine with MB	US	Frequency = 2.0 MHz ISPTA = 200 W/cm^2^ Peak negative pressure = 550~650 kPa	Decreased tumor volume Increased vascularity
[55] 2022	10 patients with breast cancer	Delivery of taxane, anthracycline, and cyclophosphamide with MB	US	Frequency = 4 MHz V-flash mode c, MI = 0.3~0.4	Improved the effects of NAC
[56] 2023	17 patients with colorectal cancer liver metastasis	Delivery of irinotecan, calcium folinate, and fluorouracil with MB	FUS	Frequency = 1.7 MHz Acoustic pressure = 0.65 MPa, DC = 0.2~0.4%	A tendency toward tumor volume reduction
**Ultrasound to Improve the Effectiveness of RT**					
**References** **Year**	**Cancer**	**Purpose**	**Type of** **Ultrasound**	**Parameters**	**Effects**
[57] 2016	Mouse breast cancer	Combination of USMB and radiation (0.2 or 8 Gy)	US	Frequency = 500 kHz Peak negative pressure = 570 kPa	Additive anti-tumor and anti-vascular effects
[58] 2017	Human hepato- cellular carcinoma, nude mouse tumor	Combination of USMB and radiation (5 Gy)	US	Frequency = 4.2 MHz Peak-negative pressure = 2.5 MPa	Increased reduction in tumor growth by USMB
[59] 2018	Human nasopharyngeal cancer, nude mouse tumor	Combination of USMB and radiation (0.2 or 8 Gy	US	Frequency = 238 kHz Acoustic pressure = 570 kPa MI = 0.8	Enhanced the effects of radiation
[60] 2020	Cancer cells (HNSCC, glioblastoma, prostate cancer)	Combination of ultrasound and radiation (10 Gy) or HT	FUS- cavitation	Frequency = 1.467 MHz Intensity = 1176 W/cm^2^	Short FUS-cavitation sensitized cancer cells to radiation and HT
[61] 2023	Human prostate cancer, nude mouse tumor	Combination of pHIFU with radiation (2 Gy)	Pulsed HIFU	Frequency = 1 MHz Intensity = 25 W (1 Hz pulse rate, 10% DC for 60 s)	Pulsd HIFU enhanced the effects of radiation
**HIFU to Shrink Tumors**					
**References Year**	**Cancer**	**Purpose**	**Type of** **Ultrasound**	**Parameters**	**Effects**
[62] 2009	Mouse HNSCC tumor	Thermal damage on SCC by continuous HIFU	HIFU	Frequency = 1 MHz Intensity = 6830.7 W/cm^2^ Acoustic pressure = 142.1 kPa	Histologically revealed necrotic area
[63] 2020	Benign thyroid nodule in patients	Treatment of tumors by thermal ablation	HIFU	Frequency = 3 MHz Maximum acoustic power = 125 W	Safe and effective to induce nodule shrinkage
[64] 2021	Mouse melanoma or breast tumor	Mechanotransductive effects by pulsed HIFU	Pulsed HIFU	Frequency = 1.15 MHz Peak negative acoustic pressure = 6 MPa, ISPTA = 2683 W/cm^2^, DC = 10%, MI = 5.6	Decreased tumor growth rates
[65] 2021	Mouse pancreatic cancer	Non-ablative pulsed HIFU in combination with ICIs	Pulsed HIFU	Frequency = 1.5 MHz Peak negative power = 17 MPa	Pulsed HIFU increased the infiltration of CD8^+^ T cells in tumors
[66] 2021	Canine oral neurilemmoma	Thermal ablation of tumor	FUS	Frequency = 1 MHz Acoustic power = 90 W DC = 50%	Complete tumor remission

MB, microbubble; DC, duty cycle; MI, mechanical index; FUS, focused ultrasound; RT, radiotherapy; ISPTA, special peak temporal average intensity; NAC, neoadjuvant chemotherapy; HT, hyperthermia; HIFU, high-intensity focused ultrasound; USMB, ultrasound with microbubbles.

**Table 3 cancers-16-02770-t003:** Previous BNCT studies using ultrasound.

References Year	Cancer	Purpose	BPA	Ultrasound	Effects
[76] 2013	Rat gliosarcoma, brain tumor model	Delivery of BPA-Fr with MRI-guided FUS in combination with MB	BPA-Fr 250 mg/kg intravenous over 2 h	558 kHz transducer Peak rarefaction pressure = 0.4 MPa A single sonication treatment duration = 20 s,	US increased the accumulation of ^10^B in the tumors
[77] 2014	Human oral cancer, nude mouse tumor	Delivery of BPA-Fr to tumors	^18^F-FBPA-Fr intravenously injected immediately after pulsed HIFU	Two min pulsed HIFU was applied to tumors	Higher tumor ^10^B in pulsed HIFU-treated mice
[78] 2014	Rat glioma, brain tumor model	Delivery of ^18^F-FBPA-Fr	Intravenous injection of ^18^F-FBPA-Fr	FUS prior to BPA administration	The tumor-to-contralateral brain ratio was 1.75-fold higher in sonicated tumors than in control tumors
[79] 2015	Human oral cancer	Delivery of BPA and BSH into SCC cells with MB	BPA, BSH 50 ppm in culture	Cells were incubated with BPA or BSH for 2 h before USMB Frequency = 1 MHz Intensity = 1 W/cm^2^ DC = 20%, 10 s	USMB increased the accumulation of BPA and BSH and BNCT decreased cell viability
[80] 2019	Rat brain tumor	Delivery of a boron polymer	Self- assembled boron- containing nanoparticles	Frequency = 1 MHz, Pressure = 0.3–0.7 MPa, DC = 0.5%, Sonication = 1 min	The T/M ratio was increased 3-fold by FUS

MRI, magnetic resonance imaging; BPA-Fr, BPA-fructose; ^18^F-FBPA-Fr, 4-borono-2-^18^F- fluoro-L-phenylalanine-fructose; USMB, ultrasound with microbubbles.

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
