# Peer review of "Ultrasound Combination to Improve the Efficacy of Current Boron Neutron Capture Therapy for Head and Neck Cancer"

_cancers, 2024, doi:10.3390/cancers16152770_

Round 1

Reviewer 1 Report

Comments and Suggestions for Authors

This paper is a review of two separate topics which the authors propose could be beneficial if brought together - something which I also find interesting and agree should be explored clinically. The two topics are the application of BNCT to patients with cancers of the head and neck which have recurred, and the application of ultrasound techniques aimed to enhance boron uptake into tumour cells and also to enhance the effectiveness of radiotherapy.

I have a number of specific comments as follows:

1. line 54, the authors make a specific point here about BPA uptake into malignant melanoma. I am not sure this specific mention is helpful (in a review of head and neck cancers) so perhaps a more general statement - line the following sentence - is more helpful. I think this text should be modified

2. Table 1 is very useful and it is good that the final column on "Clinical Outcome" contains very specific data.

3. line 143 - I have never heard the word "extrathemal" before. The cyclotron source here produces fast neutrons which are moderated in the beam shaping assembly to produce an epithermal beam which is then incident onto patients.

4. line 166. the study was with "epithermal" neutrons originating in a nuclear reactor. (Perhaps the authors are confused because the epithermal neutrons are then moderated/scattered in the patient to produce thermal neutrons that mainly trigger the reaction in 10-B.

5. The section from lines 171-179 seems to ignore the other main causes of dose to healthy tissues which are (at depth) neutron capture in Nitrogen and gamma rays from neutron capture in Hydrogen. This text should be improved.

6. The final column of Table 2 on "Effects" contains some very general statements which I would prefer to be much more specific. In some cases there is more specific text that follows table 2, but I would prefer that as much detail as possible is included in the table.

7. line 263 contains a statement that seems to me to contradict the statement on the lines before that says that application of FUS with 6 hrs did decrease tumour volumes. Please clarify / correct this text.

8. It is critical to understand whether the application of ultrasound is either (a) causing cells that previously accumulated Boron to accumulate more, or (b) causing cells that previous did not actively accumulate boron, to now do so. The authors make a comment on this on line 396 and reference 2 papers - which are both themselves reviews. This is such a key point that I would much prefer the authors to cite the original research papers and to discuss this aspect in some more detail. If cells that already take-up boron, take-up more under application of ultrasound, then the improvement in tumour control may be negligible. On the other hand if cells that previously did not take-up boron now do, then the improvement in tumour control could be very significant and important. If what happens is not clearly known then the authors should state this as this is an important topic for research.

9. I am not sure that section 6.2 is in fact useful and helpful in the context of this paper. I would prefer that this was removed or otherwise significantly reduced in length.

10. The final reference cited in the paper is previous work of these authors. The citation seems to suggest that the previously published work also mentions BNCT. On my reading it does not - so this self-citation does not seem to be appropriate. Perhaps I have missed something in which case the authors should make more clear what it is in reference 102 that is important. 

Author Response

Thank you for taking the time to review this manuscript. We appreciate your careful peer review and constructive comments.

Comment 1. line 54, the authors make a specific point here about BPA uptake into malignant melanoma. I am not sure this specific mention is helpful (in a review of head and neck cancers) so perhaps a more general statement - line the following sentence - is more helpful. I think this text should be modified

Response: Thank you for pointing this out. As indicated by the reviewer, the sentence is not always necessary in head and neck cancer. Therefore, in the revised version, we have removed the sentence “BPA is an amino acid analog that is taken up in place of tyrosine and highly taken up into malignant melanoma” (page 2, lines 56-57).

Comment 2. Table 1 is very useful and it is good that the final column on "Clinical Outcome" contains very specific data.

Response: There are variations in the data presented in each study. We have endeavored to include data common to all of these studies.

Comment 3. line 143 - I have never heard the word "extrathemal" before. The cyclotron source here produces fast neutrons which are moderated in the beam shaping assembly to produce an epithermal beam which is then incident onto patients.

Response: Thank you for your correction. We have removed “extrathermal” (page 3, line 143).

Comment 4. line 166. the study was with "epithermal" neutrons originating in a nuclear reactor. (Perhaps the authors are confused because the epithermal neutrons are then moderated/scattered in the patient to produce thermal neutrons that mainly trigger the reaction in 10-B.

Response: Thank you for your correction. We have changed “thermal neutrons” to “epithermal neutrons” (page 5, line 170).

Comment 5. The section from lines 171-179 seems to ignore the other main causes of dose to healthy tissues which are (at depth) neutron capture in Nitrogen and gamma rays from neutron capture in Hydrogen. This text should be improved.

Response: Following the suggestion of the reviewer, we have added the sentence “The other main causes of dose to healthy tissues are (at depth) neutron capture in nitrogen and gamma rays from neutron capture in hydrogen“ (page 5, lines 180-181).

Comment 6. The final column of Table 2 on "Effects" contains some very general statements which I would prefer to be much more specific. In some cases there is more specific text that follows table 2, but I would prefer that as much detail as possible is included in the table.

Response: As noted by the reviewer, there are variations in the detailed description of "effects". Basically, we have tried to follow the authors' original description.

Comment 7. line 263 contains a statement that seems to me to contradict the statement on the lines before that says that application of FUS with 6 hrs did decrease tumour volumes. Please clarify / correct this text.

Response: We believe the following part was pointed out by the reviewer. “Anti-tumor effects were not observed when FUS was performed before or immediately after drug administration [51]” (page 7, lines 259-260). This means that FUS must be performed after the nanodroplets containing paclitaxel are delivered to the tumor tissue.

Comment 8. It is critical to understand whether the application of ultrasound is either (a) causing cells that previously accumulated Boron to accumulate more, or (b) causing cells that previous did not actively accumulate boron, to now do so. The authors make a comment on this on line 396 and reference 2 papers - which are both themselves reviews. This is such a key point that I would much prefer the authors to cite the original research papers and to discuss this aspect in some more detail. If cells that already take-up boron, take-up more under application of ultrasound, then the improvement in tumour control may be negligible. On the other hand if cells that previously did not take-up boron now do, then the improvement in tumour control could be very significant and important. If what happens is not clearly known then the authors should state this as this is an important topic for research.

Response: Following the reviewer’s suggestion, we added two sentences describing the critical point and reporting on a related in vitro study. The sentences are as follows. “It is critical to understand whether the application of ultrasound causes additional boron accumulation in cells that previously accumulated boron or in cells that did not actively accumulate boron previously. Further in vitro and in vivo studies are needed to clarify this important point, but a study using cultured oral cancer cells and MBs showed that ultrasound further increased the level of boron taken up by the cells (page 12, lines 407-412).   

Comment 9. I am not sure that section 6.2 is in fact useful and helpful in the context of this paper. I would prefer that this was removed or otherwise significantly reduced in length.

Response: In sections 4.3 and 4.4, we discussed the utility of ultrasound in sensitizing cells to radiotherapy and the damaging effects of high-intensity focused ultrasound (HIFU) on cancer. We believe that applying HIFU to cancer cells as described in these sections 4.3 and 4.4 will improve the effectiveness of BNCT, which will be discussed in section 6.2. Therefore, we would like to keep section 6.2.

Comment 10. The final reference cited in the paper is previous work of these authors. The citation seems to suggest that the previously published work also mentions BNCT. On my reading it does not - so this self-citation does not seem to be appropriate. Perhaps I have missed something in which case the authors should make more clear what it is in reference 102 that is important. 

Response: Considering the suspicion of inappropriate self-citation and the need for more detailed explanations, the text containing reference 102 was deleted in the revised version. Therefore, the following sentences were deleted. “As described above, USMB stimulates ASMase activity and promotes ceramide production, leading to the radiosensitization of tumor cells and endothelial cells. Ceramide levels increase in response to RT and chemotherapy. However, when ceramide is metabolized, prosurvival factors such as S1P, ceramide-1-phosphate, and glucosylceramide are produced, reducing the antitumor effects of ceramide. Ceramide analogs and compounds targeting ceramide-and sphingosine-metabolizing enzymes exert anticancer effects. These inhibitors are also expected to function as sensitizers of BNCT for HNC [102]” (page 15, lines 515-522).

Reviewer 2 Report

Comments and Suggestions for Authors

The authors provide an overview of the use of Ultrasound in combination with Boron Neutron Therapy for Head and Neck Cancer. I am not an expert in this area, but I find the method promising and interesting. The manusscript is well written with fine tables and figures to explain the results and methods. However, the review is non-systematic and, therefore, lacks scientific methodology I can review. However, it is a fine manuscript is introducing the reader to technology and the latest developments. 

Comments

Please highlight the review is non-systematic in abstract.

What is the side-effects of treatment? Please add that to Table 1.

The manusscript is too long with many descriptions of findings from seperate studies. Please focus on essential findings and shorten the length of the manusscript. 

Comments on the Quality of English Language

Language is ok

Author Response

Thank you for your time in reviewing our manuscript. We appreciate your valuable comments. 

Comment 1. Please highlight the review is non-systematic in abstract.

Response: Thank you for pointing this out. In the abstract, we have changed the text as follows. “This review is not systematic, but outlines the current status of BPA-based BNCT and proposes plans to reduce the recurrence rate of HNC after BNCT in combination with ultrasound” (page 1, line 33).

Comment 2. What is the side-effects of treatment? Please add that to Table 1.

Response: Thank you for your suggestion. A detailed description of adverse events in Table 1 would be of little importance in this review describing a new approach to combining BNCT and ultrasound. Instead, adverse events are briefly described in section 3 (Current BPA-based BNCT Issues) (page 5, lines 163-185).

Comment 3. The manuscript is too long with many descriptions of findings from seperate studies. Please focus on essential findings and shorten the length of the manuscript. 

Response: This paper is a review of two separate topics, which we suggest would benefit from being brought together. This is because current BNCT for head and neck cancer appears to be insufficient to eradicate cancer cells by BNCT alone. To overcome this problem, we believe that the use of ultrasound in combination with BNCT is essential. Much of the research on ultrasound has been done to facilitate drug delivery or to enhance the effects of radiotherapy, but not on BNCT, a form of radiotherapy. Since no such review has been published, previous studies on BNCT and ultrasound need to be extensively described. Therefore, we would like to maintain the current length of the paper.